# Malignant cerebral infarction after ChAdOx1 nCov-19 vaccination: a catastrophic variant of vaccine-induced immune thrombotic thrombocytopenia

M. De Michele [1✉], M. Iacobucci [2], A. Chistolini [3], E. Nicolini [1], F. Pulcinelli[4], B. Cerbelli[5], E. Merenda[6], O. G. Schiavo[1], E. Sbardella[1], I. Berto[7], L. Petraglia[1,10], N. Caracciolo[7,10], M. Chiara[8], S. Truglia[9] & D. Toni[7]

Vaccine-induced thrombotic thrombocytopenia with cerebral venous thrombosis is a syndrome recently described in young adults within two weeks from the first dose of the ChAdOx1 nCoV-19 vaccine. Here we report two cases of malignant middle cerebral artery (MCA) infarct and thrombocytopenia 9-10 days following ChAdOx1 nCoV-19 vaccination. The two cases arrived in our facility around the same time but from different geographical areas, potentially excluding epidemiological links; meanwhile, no abnormality was found in the respective vaccine batches. Patient 1 was a 57-year-old woman who underwent decompressive craniectomy despite two prior, successful mechanical thrombectomies. Patient 2 was a 55-year-old woman who developed a fatal bilateral malignant MCA infarct. Both patients manifested pulmonary and portal vein thrombosis and high level of antibodies to platelet factor 4-polyanion complexes. None of the patients had ever received heparin in the past before stroke onset. Our observations of rare arterial thrombosis may contribute to assessment of possible adverse effects associated with COVID-19 vaccination.

[1] Emergency Department, Stroke Unit, Sapienza University of Rome, Rome, Italy. [2] Neuroradiology Unit, Department of Human Neurosciences, Sapienza University of Rome, Rome, Italy. [3] Hematology, Department of Translational and Precision Medicine, Sapienza University of Rome, Rome, Italy. [4] Department of Experimental Medicine, Sapienza University of Rome, Rome, Italy. [5] Department of Medico-Surgical Sciences and Biotechnologies, Sapienza University of Rome, Rome, Italy. [6] Department of Radiological, Oncological and Pathological Sciences, Sapienza University of Rome, Rome, Italy. [7] Department of Human Neurosciences, Sapienza University of Rome, Rome, Italy. [8] Neurosurgery, Department of Human Neurosciences, Sapienza University of Rome, Rome, Italy. [9] Lupus Clinic, Rheumatology Unit, Dipartimento Di Scienze Cliniche Internistiche Anestesiologiche e Cardiovascolari, Sapienza University of Rome, Rome, Italy. [10] These authors contributed equally: L. Petraglia, N. Caracciolo. ✉email: M.DeMichele@policlinicoumberto1.it

Global vaccination against severe acute respiratory syndrome-coronarovirus-2 (Sars-cov-2) has become a priority for mitigating COVID-19 pandemic. Chimpanzee Adenovirus encoding the SARS-CoV-2 Spike glycoprotein (ChAdOx1) is the Oxford-AstraZeneca vaccine (re-named Vaxzevria) authorized by the European Medicines Agency on 29 January 2021 (ref. [1]). Three recently published papers[2–4] reported case series of unusual and mostly devastating thrombosis, in particular cerebral venous thrombosis, associated with thrombocytopenia and a high level of antibodies against platelet factor 4 (PF4)-heparin, developing between 5 and 15 days from the first dose of ChAdOx1 nCoV-19 vaccine.

A vaccine-induced immune thrombotic thrombocytopenia (VITT) has been suggested to name this clinical syndrome, in order to distinguish it from the heparin-induced thrombocytopenia (HIT)[2]. Afterward, four VITT cases presenting with ischemic stroke after vaccination with ChAdOx1 nCoV-19 were reported[5,6]. A similar syndrome was also observed in some cases of young adult women (median age was 37, range 18–59 years of age) who had received the Ad26.COV2.S vaccine (Johnson & Johnson/Janssen)[7–9]. Pathophysiological mechanisms of post-vaccination PF4–polyanion antibodies induction are still unknown.

Here we describe a possible variant of the VITT characterized by a prevalent catastrophic involvement of the arterial brain district. Both patients developed thrombocytopenia with anti-PF4/polyanion antibodies, large cerebral infarct and venous pulmonary and portal vein thrombosis, within 10 days after ChAdOx1 nCoV-19 vaccination. Physicians should be aware about this rare but severe presentation of VITT, in order to promptly start the appropriate treatment.

## Results

**Patient characteristics.** Patient 1 was a 57-year-old healthy woman with a past medical history of mild hypothyroidism, in follow-up care after breast cancer treated surgically in 2012. Last mammography and breast ultrasound performed 4 months earlier were normal. She presented to our Emergency Department (ED) 9 days after the first dose of ChAdOx1 nCoV-19 vaccine and 1 h after the onset of sudden left hemiplegia, right gaze deviation, dysarthria, and left neglect, caused by right middle cerebral artery (MCA) occlusion. Since blood tests showed severe isolated thrombocytopenia (44,000 mm$^3$), the patient did not receive intravenous thrombolysis and underwent successful mechanical thrombectomy after platelet transfusion. Two hours later she underwent a second successful endovascular treatment after evidence of a repeated MCA occlusion at brain magnetic resonance (MR) and worsening of the neurological symptoms. In both procedures thrombus was collected and analyzed. Unfortunately, 12 h apart the patient developed a malignant infarct (that is an ischemia involving the whole territory of the MCA which causes space-occupying cerebral edema leading to rapid neurological deterioration)[10], due to re-occlusion of the right MCA. Decompressive craniectomy was performed on day 3 from stroke onset (Fig. 1). Intravenous betamethasone 4 mg b.i.d. was prescribed. Total body computed tomography (CT) scan showed extensive pulmonary artery and portal vein thrombosis, with no evidence of underlying malignancy. A trans-thoracic echocardiogram (TTE) revealed a structurally normal heart. Transcranial color Doppler ultrasonography with bubble test was negative for right-to-left shunt. Platelet count continued decreasing with a nadir of 23,000 mm$^3$ on day 2. Intravenous high-dose (1 g/kg) immunoglobulins (IVIG) were administered on days 4 and 5. Platelet count increased up to 344,000 mm$^3$ within 3 days, but it restarted decreasing shortly after, reaching a platelet count of 32,000 mm$^3$ 13 days after the IVIG administration (Fig. 2). Hence, plasma exchange was performed on three consecutively days (3 L exchange with 5% albumin replacement) with a low response (Fig. 2).

Treatment with fondaparinux 2.5 mg subcutaneously u.i.d. was started when platelet count had reached 50,000 mm$^3$. The patient had never been treated with heparin both in the past and recently prior to stroke. Blood gas analysis worsened on day 11. A control thorax CT scan was performed, which showed a widespread ground-glass attenuation suggestive of a severe acute respiratory distress syndrome. Patient is still hospitalized in a critical condition.

Patient 2 was a 55-year-old healthy woman with no preexisting conditions except for mild hypothyroidism. She had never been treated with heparin. Seven days after receiving the first dose of ChAdOx1 nCoV-19 vaccine she started complaining of abdominal pain and presented to a first aid on the morning of day 10th. Routine blood examinations were normal except for the high level of D-dimer (5441 ng/mL) and mild thrombocytopenia (PLT 133,000). Abdominal ultrasound was normal. In the afternoon, during her hospital stay, she experienced a transient episode of aphasia and right hemiparesis, followed 2 h later by generalized seizures and coma. Orotracheal intubation was performed. Brain CT scan, angio-CT, and perfusion CT showed occlusion of the right internal carotid artery terminus and of the left MCA, extensive ischemic cores and severe bilateral hypoperfusion, without treatable penumbra (Fig. 3).

The patient was transferred to our ED. Blood examination repeated 10 h later showed worsening of thrombocytopenia (PLT 97,000 mm$^3$) with a further decrease on the following day (PLT 59,000 mm$^3$). Treatment with IVIG (1 g/kg per day) and dexamethasone 40 mg u.i.d. were started on day 1. TTE was normal. Twelve-hour post-stroke a total body CT revealed extensive portal vein thrombosis with occlusion of the left intrahepatic branches and left lower lobe subsegmental pulmonary arteries thrombosis. Brain CT scan showed bilateral malignant MCA infarct with uncal herniation (Fig. 3). Brain death was declared 24 h later. Autopsy was not performed due to consent denial by relatives.

**Laboratory testing.** D-dimer levels were elevated in both patients at admission. Coagulation parameters were normal. No signs of hemolysis were evident.

A peripheral blood smear showed giant platelets in patient 1 and platelet anisocytosis in patient 2.

Patient 1 developed a progressive severe normochromic normocytic anemia with a nadir hemoglobin level of 5.4 g/dL on day 4 when red blood cell transfusion was performed. Patient received a second transfusion on day 14 for a subsequent reduction of hemoglobin level (nadir 7.4 g/dL).

Screening for thrombophilia was negative in both patients. Lupus anticoagulant, anticardiolipin and beta-2 glycoprotein antibodies and antinuclear antibodies tested negative.

Coagulation Factor VIII was found increased in both the patients. Factor XIII was markedly decreased in patient 1 and not dosed in patient 2. We observed increased circulating levels of von Willebrand Factor Antigen (VWF:Ag) and an increase in the levels of VWF-Ristocetin Cofactor (VWF:RCo).

**Platelet testing.** Serum of both patients showed high levels of pan antibodies (IgG, IgM, and IgA) to PF4–polyanion complexes. However, patient 1 was initially not positive but high levels of antibodies were found at day 15 (1.68 optical density—OD$_{405}$). Patient 2 resulted positive since day 1 (1.29 OD$_{405}$). Functional activity test showed that platelets from healthy donors were clearly activated by patient 1 and 2 sera in the absence of added heparin (saline), with a percent of ATP release after 20 min of

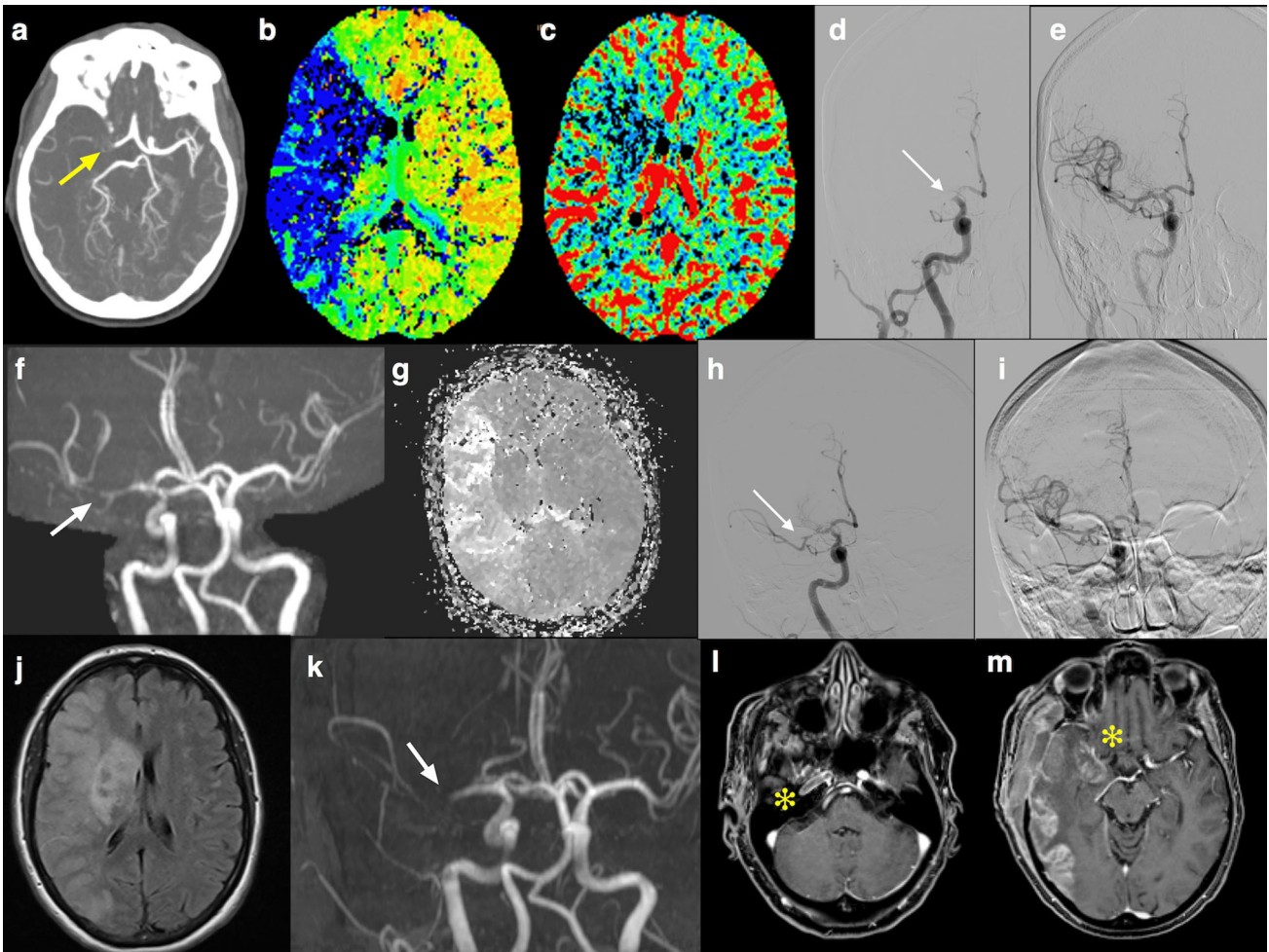

**Fig. 1 Radiological findings in Patient 1. a** Computed tomograghy angiography (CTA) showed proximal M1 segment occlusion of the right middle cerebral artery (MCA) (yellow arrow); **b**, **c** CT perfusion (CTP) maps: mean transit time (MTT) (**b**) and cerebral blood volume (CBV) (**c**) showed a large area of mismatch indicating salvageable penumbra; **d**, **e** endovascular mechanical thrombectomy achieved completed recanalization (**e**) of the MCA occlusion (white arrow in **d**); **f**, **g** MCA reocclusion on M2 segment 2 h after the procedure, as showed by 3D time of flight (TOF) magnetic resonance imaging (MRI) sequence (white arrow in **f**), with ischemic penumbra on mean transit time map in perfusion MRI (**g**); **h**, **i** second complete endovascular recanalization (**h** before and **i** after the mechanical thrombectomy); **j**, **k** fluid attenuated inversion recovery MRI at 12 h showed the extension of ischemia to superficial and deep right MCA territory with mass effect on the ventricular system (**j**), with occlusion of MCA at TOF magnetic resonance angiography (white arrow in **k**). Extension of the arterial occlusion to the right internal carotid artery (yellow asterisk in **l** and **m**).

22% and 14%, respectively. Platelet activation was not inhibited efficiently by high-dose heparin in both the patients (percent of ATP release after 20 min of 19% and 9%, respectively). Serum from healthy volunteers did not induce platelet secretion.

**COVID-19 serological testing**. Serum antibodies to spike protein of Sars-CoV-2 (IgG) were positive in patient 1 and negative in patient 2. Both the patients had negative COVID-19 real-time reverse transcriptase-polymerase chain reaction (rRT-PCR) test on nasopharyngeal swab. Patient 2 bronchoalveolar lavage (BAL) tested on day 1 was also negative.

**Clot analysis**. Clot collected from patient 1 (Supplementary Fig. 1) during the first thrombectomy was mainly composed of platelets (85% of the total material examined) and was massively infiltrated by neutrophils with scarce evidence of karyorrhexis. Histological features consistent with the presence of neutrophil DNA extracellular traps (NETs) were observed.

Clot collected during the second endovascular procedure was a red-blood-cell-rich thrombus (90% of red blood cells and 10% fibrin and platelets) with scarce neutrophils (Supplementary Fig. 2).

For detailed description of laboratory findings see Supplementary Table 1.

## Discussion

We present two cases of malignant cerebral infarct, systemic venous thrombosis and concomitant thrombocytopenia in two young healthy adult women within 10 days from vaccination against SARS-CoV-2 with ChAdOx1 nCov-19. High serum levels of antibodies to PF4–polyanion complexes were found in both patients.

Some relevant findings from these two cases need considerations.

Arterial thrombosis in VITT seems to be a rare event comparing to venous thrombosis. A population-based cohort study in Norway and Denmark described an increased rate of venous thromboembolic events but not of arterial events among recipients of ChAdOx1 nCoV-19 (ref. [11]). Greinacher et al.[2] reported 1 patient, out of 11, with aortoiliac thrombosis whereas Scully et al.[4] described two patients, out of 23, affected by MCA occlusion. In Shultz's case series no arterial thrombosis was reported[3]. More recently, Blauenfeld et al.[5] have reported the first case of VITT in Denmark who developed bilateral

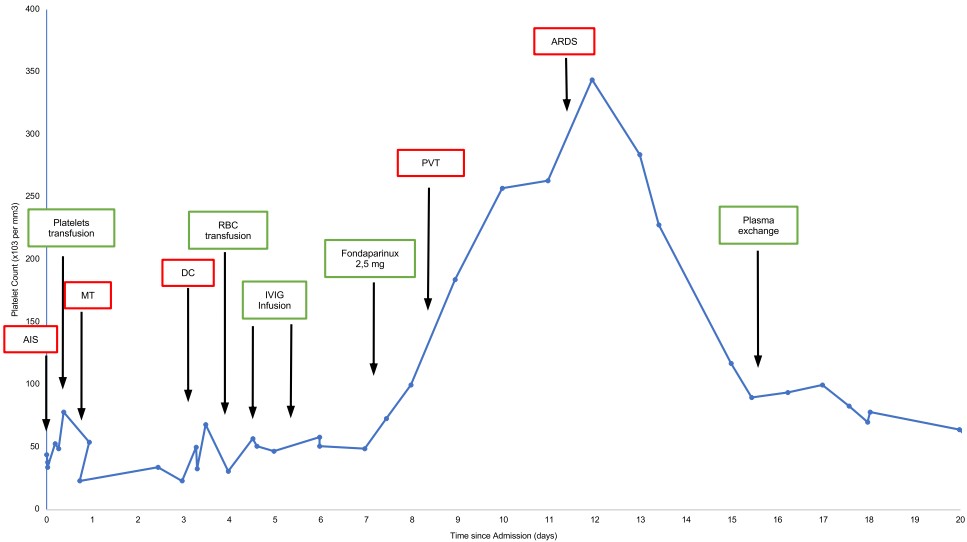

**Fig. 2 Platelet count course in Patient 1.** AIS acute ischemic stroke, MT mechanical thrombectomy, DC decompressive craniectomy, PVT portal vein thrombosis, ARDS acute respiratory distress syndrome. Source data are provided as a Source Data file.

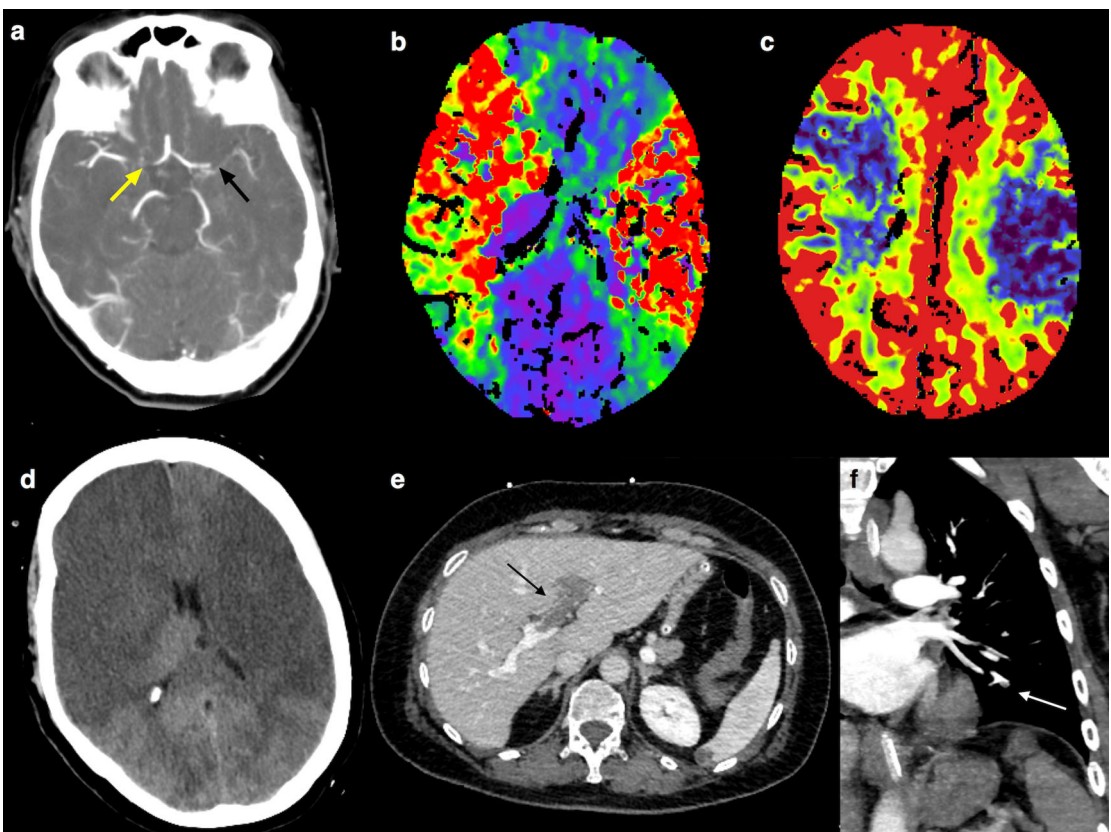

**Fig. 3 Radiological findings in Patient 2. a** Computed tomograghy angiography (CTA) showed the proximal M1 segment occlusion of the left middle cerebral artery (MCA) (black arrow) and the occlusion of the right internal carotid artery terminus (yellow arrow); **b** CT perfusion (CTP) showed bilateral infarct core (cerebral blood volume—CBV—map in **c**) and hypoperfusion (mean transit time—MTT—map in **b**), without treatable penumbra; **d** 24 h after, brain CT revealed bilateral malignant MCA infarcts; **e** portal vein thrombosis with extension to the left intrahepatic branches (black arrow); **f** subsegmental pulmonary artery thrombosis (white arrow).

adrenal hemorrhage and a fatal massive ischemic stroke treated with decompressive craniectomy, 7 days after receiving the ChAdOx1 nCoV-19 vaccination. Three other cases of VITT presenting with ischemic stroke after the administration of the ChAdOx1 nCoV-19 vaccine in United Kingdom have been described[6].

VITT appears to be similar to autoimmune HIT[2]. Range of thrombotic complications in HIT is broad with 15–20% of patients suffering arterial events, and 30–60% developing venous clots (some patients get a combination of venous and arterial thrombosis)[12]. The endothelium has been identified as a potential target for HIT antibodies. Recently, it has been demonstrated that

VWF strings released from Weibel–Palade bodies activated endothelium, are capable of binding PF4 released from activated platelets. The resulting PF4/VWF complexes forming antigen sites are recognized by HIT antibodies with consequent greater accumulation of platelets to the injured arterial endothelium, thrombus formation, and propagation[13].

Patient 1 had a right MCA stroke due to occlusion of the M1 segment. Despite two successful endovascular treatments, the patient experienced a repeated occlusion of the same vessel and developed a malignant MCA infarct. We can hypothesize that the second and the third occlusion at the same vascular site might be due to the PF4/VWF complexes recognized by HIT antibodies on the post-thrombectomy injured arterial endothelium. A high level of VWF and VWF:RCo, D-dimer, and coagulation Factor VIII support this hypothesis, suggesting a high prothrombotic state and endothelium dysfunction. Notably, at the time patient 1 was admitted, we were not yet aware of VITT, as the first paper reporting this syndrome was published a few days later. Hence, finding the low platelet count, a platelet transfusion was prescribed before the first thrombectomy, probably exacerbating the hypercoagulable state.

Histology of the first removed thrombus showed a platelet-rich clot with neutrophils, while the second one showed a red-blood-cell-rich clot. Platelet-rich thrombi are formed by VWF, NETs, and fibrin[14]. Platelet/fibrin thrombi were also found in the experimental model of HIT in veins and arteries of multiple organs[15] and HIT is also known for this reason, as "white clot syndrome"[16].

We cannot say whether histological differences of the two thrombi account for different pathophysiology. The main difference between clots was their age. Thrombus formation is a highly dynamic process: first layers of red blood cells and fibrin are main components of thrombotic material, and then platelets and white blood cells become prevalent. The second thrombus was retrieved earlier (about 2 h after its formation), so we can assume that it was fresher than the first one[17].

Pathogenesis of the severe normochromic normocytic anemia developed by patient 1 was not defined. We ruled out active bleeding through abdomen CT scan, no melena, or hematemesis were evident. Laboratory findings excluded hemolytic anemia. Bone marrow examination was not done due to the critical conditions of the patient and because of the stability of the red blood cells count after transfusion.

We found a high level of Sars-cov-2 IgG serum antibodies suggestive of a previously COVID-19 asymptomatic infection in patient 1. No Sars-CoV-2 IgG serum antibodies were present in serum of patient 2. These data would suggest that anti-spike protein antibodies might not be relevant in the pathogenesis of this syndrome. Interestingly, antibodies to PF4–polyanion complexes were initially negative in serum of patient 1, while a high antibody level was found in serum collected and tested 2 weeks later.

The serologic pattern of these two cases is similar to that of the so-called autoimmune HIT, since serum from our patients strongly activated platelets in the absence of heparin (heparin-independent platelet activation)[18]. Moreover, platelet secretion was scarcely sensitive to inhibition with high-dose heparin, as observed in two patients described by Schultz et al.[3] and in one described by Greinacher et al.[2]. A disseminated intravascular coagulation was not observed in our patients.

Both the patients had extensive arterial thrombosis of the lung and venous thrombosis of the splanchnic district. A hypercoagulable state was observed in both the patients with a high level of VWF and VWF:RCo, D-dimer, and coagulation Factor VIII. Factor XIII was markedly decreased in patient 1, probably due to excessive consumption within thrombi[19].

Treatment of this syndrome is challenging. Administration of high-dose IVIG has been suggested in an attempt to inhibit Fcγ receptor-mediated platelet activation, since in the treatment of severe autoimmune HIT this treatment resulted in rapid increase in platelet count[18,20]. Unfortunately, platelet response of patient 1 to high-dose IVIG was striking but transient. Poor platelet increase was observed after three sessions of PEx. Early non-heparin anticoagulation, in addition to high-dose IVIG and steroids, is warranted in VITT to prevent thrombosis[21,22] and, in view of the patient 1 clinical course, it should have been considered soon after thrombectomy in order to prevent reocclusion of the damaged vessel. However, early anticoagulation of patients affected by large brain infarct increases the risk of hemorrhagic transformation[23]. Moreover, the patient underwent decompressive craniectomy on day 3, which is an additional contraindication to early full-dose anticoagulation. Hence, we started with Fondaparinux 2.5 mg per day and we increased the dosage to 7.5 mg per day after 2 weeks from stroke onset.

Many scientific questions regarding VITT are still unanswered as highlighted by a recently published editorial[24]. Although limited to two patients, data collected in this report strengthen the hypothesis that platelets could be initially activated by unknown factors related to the ChAdOx1 nCov-19 vaccine/host interaction. An atypical autoimmune HIT may be the consequent event that could amplify platelet activation and coagulation cascade. The rare occurrence of VITT also in patients who received the Ad26.COV2.S vaccine (Johnson & Johnson/Janssen)[7–9] suggests a possible pathogenic role of the adenoviral vector. In addition, a potential role for the ChAdOx1 nCov-19 vaccine cell-culture-derived proteins and components (i.e. EDTA) has been recently postulated[25]. More investigations are warranted to elucidate this and other hypotheses[26] and clarify the pathogenesis of this catastrophic syndrome.

One limitation of the present paper is that we have not tested the PF4 antibody titer immediately after plasma exchange, and hence we do not know whether the recurrent thrombocytopenia was due to a persistent high level of PF4 antibody titer. Another limitation is that we have not dosed soluble DNA or citrullinated histones, markers of NETs, in the patients' sera. NETs are released from the neutrophils in response to extracellular stimuli to contain infection, but their excessive formation can be detrimental by inducing microvascular thrombosis and propagating inflammation[27]. Greinacher et al.[25] observed that VITT patient serum incubated with PF4 and purified neutrophils induces NETs formation (NETosis) with strongly enhanced response in the presence of platelets. NETosis could play a role in amplifying thrombi formation in VITT, but further studies are needed to confirm this issue.

In conclusion, our two cases of young adult women with massive brain artery thrombosis in addition to extensive systemic venous thrombosis, thrombocytopenia, and PF4–polyanion antibodies, developed within 10 days from ChAdOx1 nCov-19 vaccination, might represent a stroke variant of the recently named VITT syndrome. Ischemic stroke may be the first serious symptom of VITT. Physicians should be aware of this possibility and should carefully investigate patient medical history asking for any previous vaccination (within 4–28 days) especially in young female stroke patients. If low platelet count would be present (it could also be mild initially), a chest and abdominal CT scan should be performed to rule out venous thrombosis, together with an anti-PF4/heparin IgG enzyme immunoassay and functional platelet assay. If diagnosis of VITT is confirmed (but also if the suspect of VITT is high, without waiting for specific tests), non-heparin anticoagulation should be promptly started, if platelet count is up to $50 \times 10^9$/L, as well as therapy with high dose of steroids and IVIG. Platelet transfusion should be avoided.

## Methods

**Functional and serologic testing.** Both the patients were tested for COVID-19 through a nose/throat rRT-PCR test. A molecular test for COVID-19 on BAL was also performed for patient 2.

Serum antibodies (IgG) to the spike protein of Sars-cov-2 were measured with the Liaison-CLIA-DiaSorin system on blood samples of both the patients.

Coagulation Factors VIII and XIII, total levels of VWF:Ag, and its capability to adhere to platelet glycoprotein complex GPIb-IX-V (VWF:RCo) were dosed.

Antibodies to PF4/polyanion in serum were tested twice in patient 1 (on day 1 and on day 15) and once in patient 2 (on day 1), by using a commercial enzyme immunoassay (IgG/IgA/IgM, Immucor, Lifecodes, Waukesha, WI).

Functional test with high concentration heparin (100 IU/ml) was performed[28]. Briefly, 73 µL of washed platelet suspension was dispensed in microplates and activated with heat-inactivated (56 °C for 30 min) patient serum (27 µL) in the presence or absence of high heparin dose (100 U/mL) and left for 20 min at room temperature to allow activation. To avoid any possible thrombin effect Hirudin (5 U/ml) was added to the patient sera. ATPlite (PerkinElmer, Waltham, MA), and ATP monitoring system based on luciferin/luciferase method, (25 µL) was then added to the microplates. Emitted light was measured with a plate luminometer (Victor 3, PerkinElmer). All dosages were performed in duplicate. Effective ATP release was calculated according to the company's instructions. Platelet suitability was evaluated with U-46619 plus epinephrine ($2 + 20$ µM)-induced ATP release. Data are reported as a percentage of ATP release induced by serum/total intraplatelet ATP amount.

**Clot analysis.** Clot material from patient 1 was removed from the stent retriever during the first endovascular procedure and collected by the thromboaspiration technique during the second procedure. Histological sections obtained from the formalin-fixed paraffin-embedded material were stained with hematoxylin–eosin and phosphotungstic acid–hematoxylin. Immunohistochemistry for CD61 was also performed to highlight the platelet component.

**Informed consent.** Written informed consent was obtained from all participants according to Italian regulation, and the experimental procedure was approved by the Institutional Review Board at Sapienza University of Rome and was conducted in accordance with the Declaration of Helsinki.

**Reporting summary.** Further information on research design is available in the Nature Research Reporting Summary linked to this article.

## Data availability

The data of this report are available from the corresponding author upon reasonable requests. Source data are provided with this paper.

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

## Acknowledgements

This research did not receive a grant from funding agency.

## Author contributions

M.D.M. conceived and designed the study, enrolled the patients, interpreted the results, and prepared the original manuscript. M.I. is the neurointerventional radiologist who retrieved the thrombi of patient 1, reviewed, and edited Figs. 1 and 3 of the manuscript. A.C. is the hematologist consultant who provided treatment decision making and coagulation parameters' analysis; E.N. edited the table and Fig. 2 and critically reviewed the manuscript., O.G.S., E.S., I.B., L.P. and N.C. participated in data patients collection; B.C. and E.M. performed the histopathological examination of retrieved thrombi and edited Supplementary Fig. 1; F.P. performed the functional and serological platelets analysis and contributed to the interpretation of results; M.C., S.T. and D.T. critically reviewed and edited the manuscript.

## Competing interests

The authors declare no competing interests.
