## [Peer Review File · Nature Communications]

REVIEWERS' COMMENTS

Reviewer #1 (Remarks to the Author):

The authors have addressed this reviewer's comments.

Reviewer #2 (Remarks to the Author):

No further comments

Reviewer #3 (Remarks to the Author):

The Authors have carefully addressed the points raised by the Reviewer.
I believe the manuscript has much improved.

Minor point:

1. Please address in the discussion of the manuscript as limitations of the paper the following aspects:

- Missing PF4 testing after exchange transfusion, not available testing for soluble DNA, histones, citrullinated proteins.

This is of relevance, as it might stimulate the discussion of the research topic.

REBUTTAL LETTER.

Reviewer #3:

Minor point:

1. Please address in the discussion of the manuscript as limitations of the paper the following aspects:

- **Missing PF4 testing after exchange transfusion, not available testing for soluble DNA, histones, citrullinated proteins.**

We thank the reviewer for this suggestion. We have added in the Discussion the following sentences (pages 10-11, lines 249-258): “One limitation of the present paper is that we have not tested the PF4 antibody titer immediately after plasma exchange, and hence we do not know whether the recurrent thrombocytopenia was due to a persistent high level of PF4 antibody titer. Another limitation is that we have not dosed soluble DNA or citrullinated histones, markers of NETs, in the patients’ sera. NETs are released from the neutrophils in response to extracellular stimuli to contain infection, but their excessive formation can be detrimental by inducing microvascular thrombosis and propagating inflammation.²⁶ Greinacher et al. observed that VITT patient serum incubated with PF4 and purified neutrophils induces NETs formation (NETosis) with strongly enhanced response in the presence of platelets.²⁴ NETosis could play a role in amplifying thrombi formation in VITT, but further studies are needed to confirm this issue”.

“